# MICESE: A New Method Used for the Formulation of Key Messages from the Scientific Community for the EU Post 2020 Biodiversity Strategy

**Frédéric Gosselin [1],\* , Antonia Galanaki [2], Marie Vandewalle [3], Jiska Van Dijk [4], Liisa Varumo [5], Jorge Ventocilla [6], Allan Watt [7] and Juliette Young [7,8]**

1   INRAE, UR EFNO, Domaine des Barres, F-45290 Nogent-sur-Vernisson, France
2   School of Biology, Department of Zoology, Aristotle University of Thessaloniki,
    GR-54124 Thessaloniki, Greece; agalanaki@bio.auth.gr
3   Department of Conservation Biology, UFZ–Helmholtz Centre for Environmental Research, UFZ
    Science-Policy Expert Group, Permoserstrasse 15, 04318 Leipzig, Germany; marie.vandewalle@ufz.de
4   Norwegian Institute for Nature Research (NINA), P. O. Box 5685 Torgard, No-7485 Trondheim, Norway;
    jiska.van.dijk@nina.no
5   Finnish Environment Institute, SYKE, Latokartanonkaari 11, 00790 Helsinki, Finland;
    Liisa.Varumo@ymparisto.fi
6   Royal Belgian Institute of Natural Sciences, Museum of Natural Sciences, Vautier Street 29,
    1000 Brussels, Belgium; j.ventocilla@biodiversity.be
7   UK Centre for Ecology and Hydrology Edinburgh, Bush Estate, Penicuik EH26 0QB, UK;
    allan.watt@ceh.ac.uk (A.W.); jyo@ceh.ac.uk or Juliette.young@inrae.fr (J.Y.)
8   Agroécologie, AgroSup Dijon, INRAE, Univ. Bourgogne, Univ. Bourgogne Franche-Comté,
    F-21000 Dijon, France
\*   Correspondence: frederic.gosselin@inrae.fr; Tel.: +33-23-895-0358

**Abstract:** The European Union (EU) 2020 Biodiversity strategy will soon come to an end and may not have been as successful as envisioned. In the current context of the global biodiversity crisis, the European Commission, the research community, and broader society cannot risk another, likely ineffective, attempt by the EU to halt biodiversity loss after 2020. Through the development of the EU post 2020 Biodiversity Strategy, the scientific community of the ALTER-Net and EKLIPSE networks saw a unique opportunity to make a difference for biodiversity in Europe by better involving scientists, policy makers, and society. We developed an innovative, transparent, and collaborative process—called the multiphased, iterative, and consultative elicitation of scientific expertise (MICESE) method. This process allowed us to produce a set of 12 key messages developed by scientists for the EU to prioritize in the development of the new post 2020 biodiversity strategy. These key messages were structured according to their systemic value, scale, and nature. We provide insights and analyses of the new MICESE method before reflecting on how to improve the future involvement of scientists in science–policy interfaces.

**Keywords:** science–policy interface; European Union; consultation process; methodology; elicitation of scientific expertise; iterative

## 1. Introduction

The current European Union (EU) Biodiversity Strategy to 2020 will soon come to an end [1]. Its aim has been "to halt the loss of biodiversity and improve the state of Europe's species, habitats, ecosystems and the services they provide, while stepping up the EU's contribution to averting global biodiversity loss" (p. 7). Progress towards implementing the actions and achieving the six targets

set out in the Strategy, however, has been poor. The mid-term review by the European Environment Agency (EEA) [2] demonstrates that although actions on the ground, supported by adequate financing, can protect and restore nature and the benefits it provides to people on a European scale, we are unequivocally falling short on all but one of the targets. The Intergovernmental Science–Policy Platform on Biodiversity and Ecosystem Services (IPBES) assessment on the state of biodiversity and ecosystems in the European region, published in 2018, reported that although some progress has been made, trends remain negative overall, with potential consequences for the economy and society [3]. The consequences of a continued failure to address the decline in biodiversity and ecosystem services both in Europe and globally is now widely recognized, notably through the UN Agenda 2030 and sustainable development goals (SDGs), which highlight the crucial role of biodiversity in reaching the 17 SDGs as well as the importance of scientific methods and tools for their implementation [4–6].

Despite the failure to meet the 2010 target and the pessimistic prognosis in meeting the 2020 target, biodiversity loss remains a major societal concern. In 2015 and 2019, the European Commission's Special Eurobarometer on "Attitudes of Europeans towards Biodiversity" revealed an important increase in the awareness of Europeans on the importance of biodiversity for people [7,8]. Both reports found that over 75% of Europeans totally agreed that "we have a responsibility to look after nature", while there was a strong increase in the proportion of Europeans that totally agreed that: "biodiversity and healthy nature are important for our long-term economic development" (+6%, or 62%) and "biodiversity is indispensable for the production of goods such as food, materials and medicine" (+8%, or 61%). In 2015, 49% thought they were or would be personally affected by biodiversity loss and more than 80% of European citizens considered the various effects of biodiversity loss to be a serious issue at the global and European levels. The need for action at the European scale remains clear: the 2019 Eurobarometer reported that "most citizens see the EU as a legitimate level to take action on biodiversity and ecosystem services" [8].

At a time when there is both decreased influence of objective facts in shaping public opinion and increased demand for more evidence-informed policy decisions on environment-related issues [9–11], it is crucial that science and scientists find different ways to organize themselves and develop adapted methods to provide timely policy relevant evidence. ALTER-Net (www.alter-net.info) and EKLIPSE (www.eklipse-mechanism.eu/) jointly saw in the development of the post 2020 EU Biodiversity Strategy (hereafter called p2020EUBS) a unique opportunity to join forces and take advantage of their individual strengths to make a difference for biodiversity in Europe by engaging scientists, policy makers, and society. The aims of this engagement were to (1) stimulate both the European Commission and the research community toward a robust and effective p2020EUBS and (2) develop an engagement strategy to create awareness and ownership from science, policy, and society to implement it.

In this paper, we describe the process we used to gather input from the scientific community to elaborate what we call scientific key messages (KMs) for the p2020EUBS. We call this method the multiphased, iterative, and consultative elicitation of scientific expertise (MICESE) method. By reflecting on this method, we analyze what type of messages a multiphased consultation process produced and how they developed, and we identify the advantages and disadvantages of MICESE compared to other existing approaches. Finally, we consider the lessons learnt for improved involvement of researchers in science–policy interfaces.

## 2. Materials and Methods

During the winter of 2017, ALTER-Net and EKLIPSE developed the idea of organizing a joint horizon scanning workshop to discuss how the EU biodiversity strategy should develop after 2020, how to engage scientists in this process, and how to shape ALTER-Net's biennial conference around this theme. The horizon scanning workshop was organized in Peyresq, southern France, in September 2018. It included scientists and other participants with broad knowledge of how EU policy is developed and experience of science–policy–society interactions. This mix of participants focused the discussion towards concise, policy-relevant outputs for decision-makers. Around 15 people attended

the workshop. The decision was taken to continue the work in two parallel but related processes that would aim to provide relevant input to the p2020EUBS: one scientific process and one societal process. This paper reflects on the scientific process. For the description and analysis of the societal key messages results, please refer to [12].

The scientific process was organized using the ALTER-Net and EKILPSE conference held in June 2019 as an important milestone of the process. We devised a simple process to propose—and gradually select—KMs from the scientific community to the EU for the p2020EUBS. The aim was to have a dynamic process, where the KMs were elicited through reactions of scientists—this meant that the KMs could change, or become redundant and disappear, and that new KMs could emerge as part of the consultative process. The consultative process began in January 2019 and a call for engagement in both the scientific and societal processes was broadcast by both ALTER-Net and EKLIPSE.

The consultation was organized in several phases. The first phase involved building an editorial team (now authors of this paper plus Thibault Datry at the beginning of the process), outlining initial KMs from the Peyresq workshop, and devising a consultation method for the KMs. We started with nine KMs resulting from the Peyresq workshop. The aim of the second phase (January–June 2019) was to gather expert judgements [13] on KMs for the p2020EUBS. During this phase, the scientific community was consulted through a permanently available survey on the internet—using the site old.soorvey.com—to get their input on: (i) the scientific soundness, policy relevance, and any other comments on existing KMs and (ii) proposals for new KMs. In terms of policy relevance, the question asked was: "I agree this key message is important to take into account in EU biodiversity post 2020", with a score between 0 and 10—10 being the highest relevance. Regarding the scientific basis, the participants to the survey were asked about the "Level of scientific knowledge substantiating the key message" with 6 possible levels: "Well established", "Partially established", "Mixed (knowledge pro and con", "contra (knowledge mostly against the key message)", "Unknown scientifically", "I don't know". Defining expertise was discussed among the editorial board on two occasions, mainly in relation to who qualified as an expert. Each KM was attributed to one member of the editorial team, who read the comments made on the KM, and proposed responses and modifications of the KM based on these comments. The responses and modifications were presented and discussed during editorial video meetings. After the presentation of the proposed changes by the person responsible for the KM, there was a discussion and a decision made—mostly by consensus. Our objective was to keep the number of active KMs to a minimum (less than 20), to avoid a situation where there would have been too many messages that would have ceased to be "key". During this phase, a strong emphasis was put on transparency of the process by writing and sharing the minutes of our meetings and by saving the files that explained all the decisions made on each KM (see an example in Supplementary Material File S1). To enable us to judge them objectively, the contributions from the scientific community were kept anonymous throughout, except for the first author (FG) who managed the outputs of the online consultation. All these procedures were written in a protocol that was regularly updated (see Supplementary Material File S2).

Following the consultation process described above, 17 draft KMs were presented to the 130 delegates of the biennial ALTER-Net and EKLIPSE conference held in Ghent 17–19th June 2019. The overarching theme of the conference was "The EU Biodiversity Strategy Beyond 2020" and a major aim of the conference was to facilitate insights and discussion on the draft KMs as further input to their development.

After a plenary presentation introducing the delegates to the KM drafting process, they were invited to post written comments on the draft KMs during the first two days of the conference. Prior to the conference, all plenary and workshop chairs were asked to remind the delegates of the opportunity to both comment on the existing KMs and add new ones, particularly in relation to the themes of the specific workshop sessions. The KMs were also presented during the workshop sessions, and workshop chairs were encouraged to pick up suggestions from the workshop participants on the KMs.

The editing team considered this input and edited the KMs accordingly on the morning of the third day of the conference.

In a final plenary session, the redrafted set of 16 KMs were presented for discussion. The conference delegates were asked for further comment using the mentimeter software (https://www.mentimeter.com/—an interactive software that allows the audience to use their smartphones to connect to the presentation where they can answer questions and give feedback. The answers are visualized in real time and can be analyzed more thoroughly after the presentation ends). In terms of feedback, participants were also asked whether each KM should be retained and/or combined with other KMs. This input was used for the next step in the process.

The last phase of the scientific process on the KMs was held from July to December 2019. We decided not to have a further external consultation or survey, although this was originally planned, in order to respect the decisions and input from the participants at the Ghent conference. We agreed on a uniform structure for all KMs, consisting of a KM number and version (for internal use only), title, KM summary in one sentence, a rationale for the KM, and some implications of the KM in bullet points. The links with the current 2020 EU biodiversity strategy were also included.

A final review of the KMs was done internally by the editorial team, by assigning new reviewers to each KMs (i.e., 2–3 members of the editorial team for each KM) to minimize any bias of the original person(s) in charge. After having developed their comments independently, reviewers in charge of re-reviewing the KMs prepared a written document of their reviews, highlighting the links of the KM with other KMs. They then worked with the editor responsible for each KM, agreeing on a final version of the KM.

At the end of editing process, the KMs were classified under broad themes [14]. We decided to base the classification on the policy relevance of KMs as judged by the persons that participated in the survey phase. This allowed us to highlight a first group of systemic KMs—i.e., messages that had broad relevance and applicability and that could have significant impact on the European socio-ecosystem outside their domain, a second group of important KMs but of lower systemic scope, and a third group of KMs with narrower relevance. We also set one KM that was of global scope, as clearly different from the others.

This process resulted in us developing and applying a multiphased, iterative, and consultative elicitation of scientific expertise (MICESE) approach summarized in Figure 1.

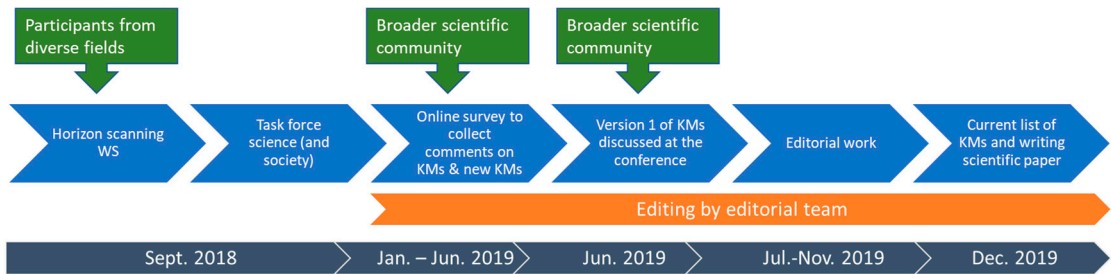

**Figure 1.** Schematic timeline representation of the MICESE process. MICESE stands for multiphased, iterative, and consultative elicitation of scientific expertise.

## 3. Results

### 3.1. Reporting the MICESE Process

From nine KMs proposed initially, our process led to double this number (18 KMs) being considered. Eight new KMs were proposed through the internet survey and one during the Ghent conference. One KM was rejected before the Ghent conference—because it referred to geodiversity that we perceived had no link through public policies and societal changes with biodiversity—and five were merged with other KMs due to a high overlap between them, to give a final list of 12 KMs

(see Section 3.2). Not counting the final reviewing process, each KM was revised approximately three times, resulting in an average of 3.3 versions per KM before the Ghent conference.

We held 23 web video meetings during the whole process, 4 meetings to agree the initial set of draft KMs before launching the online consultation, 12 during the online consultation, and 7 after the Ghent conference. These meetings lasted 1h 30 min on average.

We received 393 comments and/or scores (from 44 contributions on one or more KMs) during the online survey phase, and 90 comments on individual KMs during the Ghent conference, with most comments (16) received on the KM on monitoring. The disciplinary profile of participants to the internet survey (81% in Ecology, 26% in Environmental Sciences, 9% in Humanities, and 7% in Mathematics/Physics—with possible double affiliations) showed a strong input from scientists from the ecological field (more than 80%), followed by environmental sciences (around one-quarter). Less than 10% of the participants had a profile in humanities (including economics and social sciences). Thirty-two out of 44 had a PhD.

Of the 18 KMs that were reviewed in the online survey phase: (a) eight were merged, resulting in four new KMs (KM2, KM4, KM6, KM9), (b) eight KMs remained (KM1, KM3, KM5, KM7, KM8, KM10, KM11, KM12), and (c) two KMs were deleted (former KM16 and KM18), based on the votes and comments of the delegates at the Ghent conference (see Table 1). See Supplementary File S3 for the correspondence between both sets of KMs.

**Table 1.** Votes (%) of the Ghent mentimeter poll (online poll software) on the key messages (KMs). Ghent ALTER-Net and EKLIPSE conference attendees were asked whether we should keep each KM in our final list. Possible—exclusive—answers were: "Yes, as such", "Yes with rewording", "No fuse it with another key message", "No, delete it". In each column, we kept the maximum score in case KMs were merged.

| Key Message | Yes (%) | Yes, as Such (%) |
|---|---|---|
| KM1 | 87 | 28 |
| KM2 | 91 | 41 |
| KM3 | 89 | 22 |
| KM4 | 94 | 13 |
| KM5 | 95 | 22 |
| KM6 | 83 | 28 |
| KM7 | 97 | 33 |
| KM8 | 67 | 21 |
| KM9 | 78 | 37 |
| KM10 | 94 | 35 |
| KM11 | 62 | 17 |
| KM12 | 64 | 44 |

*3.2. Structured List of KEY Messages*

The KMs are presented here in relation to their perceived policy relevance according to the outcome of the online survey (see Table 2):

(a) Overarching, systemic messages—broad and of most importance: KM2 to KM6, which received policy relevance scores above 8.40;

(b) Important KMs but of narrower scope (less systemic): KM1, KM7 to KM9;

(c) Less important or much narrower KMs: KM10 to KM12, which received the lowest policy-relevance scores (less than 7.84; except for KM12, which received only a few scores) and were judged as either narrower, less systemic, or simply of lower policy importance by the editorial team.

**Table 2.** Votes (%) of the online consultation phase on the policy relevance and scientific basis of the key messages (KMs). In terms of policy relevance, the question was: "I agree this key message is important to take into account in EU biodiversity post 2020", with a score between 0 and 10. Regarding the scientific basis, the participants to the survey were asked about the "level of scientific knowledge substantiating the key message" with 6 possible levels: "Well established", "Partially established", "Mixed (knowledge pro and con", "contra (knowledge mostly against the key message)", "Unknown scientifically", "I don't know". We here give the frequency of "Well established" answers. If the final KM resulted from two merged KMs, we kept the maximum score from the separate KMs.

| Key Message | Mean Policy Relevance Score (Between 0 and 10) | Scientifically "Well Established" (%) |
|---|---|---|
| KM1 | 7.84 | 19 |
| KM2 | 8.76 | 41 |
| KM3 | 8.81 | 43 |
| KM4 | 8.53 | 42 |
| KM5 | 8.87 | 61 |
| KM6 | 8.47 | 50 |
| KM7 | 8.18 | 43 |
| KM8 | 7.90 | 43 |
| KM9 | 8.15 | 45 |
| KM10 | 7.82 | 39 |
| KM11 | 7.10 | 25 |
| KM12 | NA | NA |

KM1 is listed first because of its global relevance (see Figure 2).

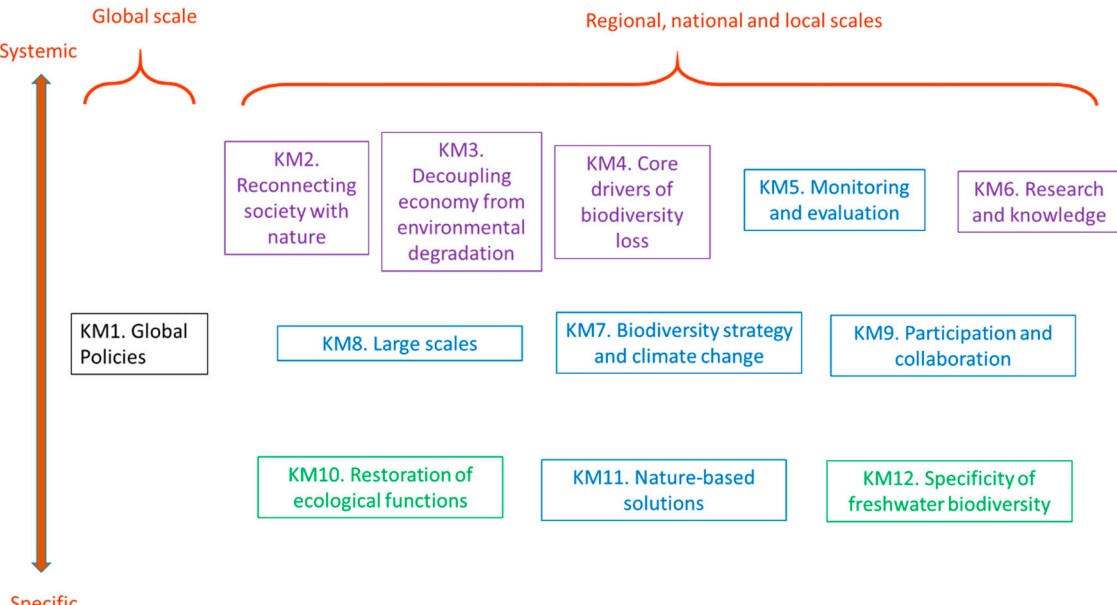

**Figure 2.** Schematic representation of our 12 key messages (KMs), along a scale (horizontal) axis and a systemic vs. specific (vertical) axis, and according to the nature of the KM (color). KMs in purple represent KMs that can be considered as drivers of the system. Green ones are more related to the state of biodiversity and blue ones to responses of society to the biodiversity issue. Some KMs could have a mixed classification in these respects (e.g., KM10), but we kept the classification that we judged was dominating.

### 3.2.1. KM1. European and Global Policies

*Europe should take a leading role in establishing an improved global policy on biodiversity.*

European policy on biodiversity has—in principle, at least—been based on the recognition that biodiversity and the drivers affecting it have to be dealt with not only at the European scale, but also globally, recognizing the impact of international decisions on European biodiversity and vice versa. In the future, therefore:

- The development of European policy on biodiversity, whilst addressing the limitations of current policy including the perverse effects of European policy elsewhere in the world, must continue to be done in close collaboration with, and building on, the goals of the UN Convention on Biological Diversity (CBD), the UN 2030 Agenda for Sustainable Development, and its SDGs and the work of the IPBES.
- The European Union should not simply follow the example of global policy developments but take an ambitious lead on biodiversity policy. It should learn from best practice everywhere, and not limit itself due to lack of vision and ambition.

Link with EU Biodiversity strategy: Target 6, Help stop the loss of global biodiversity. http://ec.europa.eu/environment/nature/biodiversity/strategy/target6/index_en.htm.

### 3.2.2. KM2. Reconnecting Society and People with Nature for an Improved Recognition of the Value of Biodiversity and Ecosystem Services

*The EU should develop initiatives that reconnect people and nature with the potential outcome of encouraging societal and individual mind-sets that recognize human dependence on nature, and the multiple values of biodiversity.*

Reversing the ongoing biodiversity decline is hampered by the disconnect between humans and nature. Urbanization, buying food directly in supermarkets, and having jobs and daily life not in direct contact with nature distance people from biodiversity; this manifests itself in our misunderstanding of the intrinsic and relational values of biodiversity and of our dependency on biodiversity. Biodiversity and ecosystem services are the foundation of all potential human activities and quality of life and should be managed and shared in a sustainable and equitable way rather than as means to sectorial goals that benefit a small minority of people. This requires that conservation of biodiversity and ecosystem services go hand in hand with the sustainable use of natural resources. To enable the reversal of the ongoing biodiversity loss and ecosystem services degradation, a change in mindsets is needed across society to reconnect society and people with nature.

The EU strategy should include measures such as:

- Focusing on environmental respect, awareness of human dependence on biodiversity, awareness of limits to growth, and the advantages of circular economy.
- Changing the economic and governance approach so that biodiversity conservation is no longer a minor consideration but protected by regulatory measures across all sectors.
- Integrating versatile education on biodiversity, ecology, and sustainability in primary and secondary school curriculums for improved understanding of the importance and dependence of biodiversity and encouraging citizen science activities in education to enable contact and experience with nature.
- Preserving existing or facilitating the creation of new collective and individual relational values with nature.
- Stressing the importance of nature for human mental and physical wellbeing.

For these actions to have impact, the EU will need to improve the coherence of the ethical foundations of its policies with the recognition of human dependence on nature and of the intrinsic and relational values of biodiversity.

Link with EU Biodiversity strategy: Target 2, Maintain and restore ecosystems, Target 3, Achieve more sustainable agriculture and forestry, and Target 4, Make fishing more sustainable and seas

healthier and Target 6, Help stop the loss of global biodiversity.  http://ec.europa.eu/environment/nature/biodiversity/strategy/index_en.htm#stra.

### 3.2.3. KM3. Decoupling Economic Development from Environmental Degradation

*The new EU biodiversity strategy should stimulate policies that decouple economic development from environmental degradation, promoting such development within the ecological limits of the planet and the UN Agenda 2030 and its Sustainable Development Goals.*

Economic growth, as measured through traditional gross domestic product (GDP) across Europe, has indirectly reinforced drivers of biodiversity and ecosystem services loss.  Although a range of policies, including environmental taxation, have been implemented to decouple economic development from drivers of biodiversity loss, policy instruments such as harmful agricultural and fishing subsidies, still impede transitions towards a sustainable future.  Recommendations for decoupling economic development from environmental degradation include:

- Transforming taxation and other policies across Europe.
- Developing and using new indicators that incorporate human wellbeing, environmental quality, employment and equity, biodiversity conservation, and nature's ability to contribute to human wellbeing.  Existing problematical metrics such as GDP could, for example, be replaced by a natural capital accounting approach and quality of life indicators such as a happiness index.
- Enhance other perspectives such as nature-based solutions and the circular economy.
- Promote biodiversity mainstreaming into EU policies that include economic aspects, such as the *Europe 2020 Strategy for Smart, Sustainable, and Inclusive Growth* and *Towards a sustainable Europe by 2030*.

Link with EU Biodiversity strategy:  New target.

### 3.2.4. KM4. Core Drivers of Biodiversity Loss and Integration Across Sectors

*The new EU biodiversity strategy should target enhanced mainstreaming of biodiversity into other sectoral policies because the direct drivers of biodiversity loss are the consequence of indirect, or core, drivers such as human population density and the consumption of resources.  This should include better integration across sectors and the designing of comprehensive biodiversity policy mixes.*

The direct drivers of biodiversity loss include climate change; overexploitation of organisms through, in particular, hunting and fishing; land use change such as intensification of agriculture and forestry; pollution; and invasive species. These are the consequence of indirect, or core, drivers such as human population density and unsustainable resource consumption. Recommendations for enhancing mainstreaming of biodiversity into policy sectors include:

- Greater recognition of the relationships between human activity and biodiversity across all policy sectors, thereby transforming all relevant policies, for example, the identification and elimination of harmful subsidies in the common agricultural policy (CAP);
- Taking trade-offs involving biodiversity and ecosystem services between different policy and economic sectors (including agriculture, urban planning, water use and fisheries sectors, and forestry) into account, for example the need to consider the impact on biodiversity of renewable energy policy implemented through bioenergy fuelwood;
- Designing comprehensive biodiversity policy mixes, including but not limited to regulation; and

  o integrating appropriate indicators that make a portion of funding conditional on ecological performance, e.g., habitat quality or management outcomes;
  o measuring national welfare using economic indicators that take into account the diverse values of nature;

    o    developing fiscal reforms to provide integrated incentives and provide leverage to redirect activities that support sustainable development;

    o    supporting mainstreaming of biodiversity in the UN 2030 agenda for sustainable development and its SDGs.

Link with EU Biodiversity strategy: New Target.

### 3.2.5. KM5. Monitoring and Evaluation

*The new EU biodiversity strategy should recognize the unique place of monitoring trends in biodiversity and ecosystem services, not only for the implementation of existing policy, but also to provide early warnings if new action is necessary and to guide the development of new policies across all relevant sectors. Future monitoring of biodiversity should pay more attention to those species, habitats, and biogeographical areas that have been relatively neglected. Scientific knowledge and expertise, particularly taxonomic, are needed to support monitoring and citizen science should play a greater role.*

Monitoring trends in biodiversity and ecosystem services are necessary for: The implementation of policy, for example by assessing progress towards policy targets; evaluating the effectiveness of specific policies; informing the development of new nature conservation policies; providing early warnings to enable where and when action is needed; supporting adaptive management; and enabling the mainstreaming of biodiversity in other policy sectors. Although many species and habitats are already monitored, recent evidence of declines in insect abundance, for example, have shown that not all taxa are adequately monitored, including those assumed not to be endangered. Thus, for improved monitoring:

- There is a need to ensure adequate coverage across all taxa and biogeographical areas and include sufficient data in terms of quantity and quality to allow vigorous evaluation of policies such as the Habitats Directive and Natura 2000.
- LIFE and other EU projects could be incorporated in monitoring programs because of their data on the conservation status of species, although there needs to be recognition that not all member states have sufficient resources and the data from some countries may not be up to date.
- Monitoring of biodiversity should be adequately supported by experts, including taxonomists, and the latest developments in species identification.
- Monitoring should address status and trends in ecosystems, species, functional, and genetic diversity.
- Volunteer citizen scientists can play a major role in monitoring, but this requires incentives and support mechanisms for the collection, sharing, and analysis of data.
- The need to adequately share data should be addressed through, for example, funding for database construction.
- Equally important is the need for collecting and synthesizing social science data along with environmental data, including data on drivers of biodiversity change, from agriculture, energy, transport, and other sectors, in order to produce knowledge useful for developing, implementing, and evaluating policies and practices related to the conservation of biodiversity and the sustainable use of ecosystem services.
- Monitoring should also include societal attitude, and effectiveness of education related to biodiversity.

Link with EU Biodiversity strategy: Target 1, Protect species and habitats, particularly Action 4, Make the monitoring and reporting of the EU nature law more consistent, relevant and up-to-date http://ec.europa.eu/environment/nature/biodiversity/strategy/target1/index_en.htm.

### 3.2.6. KM6. Research and Knowledge-Informed Decision-Making and Implementation

*The new EU Biodiversity Strategy should engage/initiate institutional mechanisms that can effectively synthesize scientific and other types of knowledge and suggest ways in which policymakers and other societal actors can actively incorporate that knowledge into actions that promote biodiversity and ecosystem services.*

For policy needs, development, implementation, and assessment. we recommend an effective knowledge strategy that promotes knowledge synthesis and development of policy options to be included in the strategy. Such an approach should:

- Be informed by the best available knowledge, coming from wide-ranging disciplinary research;
- Use interdisciplinary and transdisciplinary approaches, through for example the engagement of citizens, practitioners, other relevant societal actors;
- Include local and indigenous knowledge, as highlighted in the assessments of the IPBES;
- Make use of mechanisms that support knowledge-informed decision-making, such as the EKLIPSE mechanism.

Given the complexity of biodiversity and ecosystem services decline, it is crucial all relevant research disciplines develop and cooperate to gain insight in the interactions of all factors. Using transdisciplinary research to improve the insights in these interactions is key to achieve sustainable development but should be treated with caution to avoid promoting individual or organizational interests.

To address the challenges of current institutional settings, research and action should also focus on constructive ways in which policymakers can and will actively incorporate knowledge into actions to address biodiversity loss.

Link with EU Biodiversity strategy: Knowledge and Data: http://ec.europa.eu/environment/nature/knowledge/index_en.htm.

### 3.2.7. KM7. Biodiversity Strategy/Climate Change

*The new EU Biodiversity Strategy should systemically address the impacts of climate change on biodiversity.*

The 2020 Biodiversity strategy and previous EU Nature policies (esp. the Habitats Directive) have been developed with limited consideration of climate change. With increasing awareness of climate change and its consequences for biodiversity, climate proofing of the new EU biodiversity strategy is urgent. In the specific case of the Habitats Directive, this has partly been taken into account by the EC through its guidelines on Climate Change and Natura 2000, but more should be done, such as:

- Jointly rethinking different actions (e.g., Habitats Directive jointly with the Green Infrastructure strategy, or the Habitats and Birds Directives and other conservation policies in Europe) in light of climate change with an emphasis on connectivity;
- Revising the Habitats Directive to allow flexibility in the selection, management, and setting targets within sites in light of climate change;
- Establishing the Natura 2000 network more coherently across boundaries instead of establishing it independently within each EU member state;
- Enlarging the conservation toolbox, considering connectivity between protected sites; protecting non-pristine sites with high ecological potential in the context of climate change; using disturbance for some sites for specific biodiversity goals while also considering using management cessation as a management tool in other sites.

Link with EU Biodiversity strategy: Whole strategy, with a specific focus on: Target 1, Protect species and habitats, particularly Action 1: Complete the Natura 2000 network and ensure its good management http://ec.europa.eu/environment/nature/biodiversity/strategy/target1/index_en.htm.

3.2.8. KM8. Incorporate Regional/Transnational Processes and Long-Term Temporal Scales to Enhance Success and Efficiency of Biodiversity Policy

*The new EU Biodiversity Strategy should give greater consideration of large temporal and spatial scales and processes to enhance its success and efficiency.*

Policy (e.g., Habitats and Birds Directives, Water Framework Directive, CAP, etc.) and management (e.g., monitoring, restoration, conservation) should incorporate larger temporal and spatial scales and processes (including transnational aspects when relevant) to enhance success and efficiency of conservation and restoration of biodiversity and ecosystem services. Indeed, biodiversity changes in one system affect biodiversity and processes in other systems.

Potential actions could include:

- Long-term considerations: Use historical baselines within policy and management; integrate the very likely occurrence of long-term time lags between action and results on biodiversity, particularly in monitoring and evaluation of policy and management;
- Large spatial scales: Support nations to work more together trans-boundary; integrate large-scale processes and pressures in the framing of policy.
- Monitoring and indicators: Introduce the perspective of large temporal and spatial scales in monitoring methods and indicators across the EU so that processes operating at such scales are taken into account.

Link with EU Biodiversity strategy: Target 2 Maintain and restore ecosystems, Target 6 Help stop the loss of global biodiversity, http://ec.europa.eu/environment/nature/biodiversity/strategy/index_en.htm#stra.

3.2.9. KM9. Participation and Collaboration for Ethical and Sustainable Decision-Making

*The new EU Biodiversity Strategy should provide the ethical foundations for decision-making across generations and emphasize participation of diverse stakeholders to foster the integration of various forms of knowledge in policy- and decision-making while promoting shared responsibility.*

The new Biodiversity Strategy should integrate various forms of knowledge through participatory processes and explicitly encourage the engagement of younger generations to safeguard the foundations of decision-making across generations. Currently, youth worldwide protest against climate and environmental change and as new biodiversity policies will affect the future generations, their perceptions should be systematically acknowledged. Practitioners, policy, and science should work multilaterally for sustainable decision-making and management. Increasing involvement of stakeholders will aid transparency and help integrate various forms of knowledge in policy- and decision-making, which is central for transformative change and promoting shared responsibility of biodiversity and decision-making related to it.

Recommendations in this regard include the need to:

- Involve and reach the younger generations through diverse methods of communication to enable their meaningful participation in decision-making and building the ethical foundation of future policymaking.
- Develop and implement systematic and flexible modes of participatory processes for different audiences in decision-making and knowledge production to support policy, including opportunities to participate via citizen science.
- Promote effective existing and new channels of transparent and legitimate participation and multilateral communication between different stakeholders.

Link with EU Biodiversity strategy: New target.

### 3.2.10. KM 10. Restoration of Ecological Functions

*The new EU Biodiversity strategy should target restoration of ecological functions as an important tool to conserve biodiversity and ecosystem services, including those involved in nature-based solutions.*

Restoration of ecological functions provide an important nexus for the management of ecosystems, acknowledged by Target 2 of the current EU Biodiversity Strategy and the 2021–2030 UN decade on Restoration. Yet, ecological functions are relatively absent from the current Biodiversity Strategy, even though they are implicitly present in the Habitats Directive through the notion of good conservation status of habitats. Specific recommendations to focus on ecological function in the new strategy include:

- Defining the priority ecological functions to be targeted, by focusing on functions that both affect and are affected by biodiversity and ecosystem services. Priorities should also include functions that both contribute to the Biodiversity strategy and the Climate change strategy, and could, for example, include climate resilience and water management, including wetlands.
- Defining research priorities in this field, with a focus on the link between biodiversity and ecosystem services through ecological functions and subsequently on the refinement of the notion of healthy ecosystems.
- Incorporating the assessment of ecological processes in biodiversity monitoring, thereby providing a better understanding of the functional health and ecological vitality of ecosystems.
- Addressing constraints to ecological restoration, including perverse subsidies, complex legal and institutional arrangements, and low local political will.

Link with EU Biodiversity strategy: Target 2 http://ec.europa.eu/environment/nature/biodiversity/strategy/target2/index_en.htm.

### 3.2.11. KM11. Nature-Based Solutions (NbS) for Sustainable Development and Nature Conservation

*The new EU Biodiversity strategy should promote the use of NbS whenever possible, to better integrate socio-economic, conservation, and other environmental objectives.*

Nature-based solutions (NbS) may be defined as solutions to societal challenges that are inspired and supported by nature, which are cost-effective, simultaneously provide environmental, social, and economic benefits and help build resilience (https://bit.ly/2TUNsxc). NbSs for development projects seek to integrate socio-economic, environmental, and ecological aspects and values. This improves the acceptance and effectiveness of these initiatives, especially locally, while solving societal and environmental, including biodiversity, challenges.

Recommendations on how to mainstream NbS in the context of biodiversity conservation include:

- Explore and aid the upscaling of local small-size NbS and encourage NbS mixes.
- Recognize the circumstances under which the use of NbS are most beneficial and avoid, in particular, undermining other nature conservation measures with NbS.
- Monitoring and evaluation of NbS should be included in the design of the initiatives and follow, for example, the recommendations of the Monitoring and Evaluation for Ecosystem Management MEEM framework (https://bit.ly/2IPC8Mm).
- Encourage and educate local communities on NbS to illustrate the positive synergies of them for biodiversity and local development.

Link with EU Biodiversity strategy: New target.
More on nature-based solutions: https://ec.europa.eu/research/environment/index.cfm?pg=nbs.

### 3.2.12. KM 12. Specificity of Freshwater Biodiversity

*The new EU biodiversity strategy should explicitly address the conservation and sustainable use of inland water biodiversity.*

Freshwater ecosystems are a unique and important component of global biodiversity, providing clean water, food, livelihoods, and many other ecosystem services. At the same time, inland waters face distinct threats so that the biodiversity of rivers, lakes, and inland wetlands are declining at rates that far exceed those seen in forests and oceans, with an urgent need for action to reverse this trend. Yet, inland water lack explicit recognition in the biodiversity strategy so far (other than e.g., marine systems, or agricultural and forestry). Furthermore, coherence and complementary between biodiversity directives and other directives such as the Water Framework Directive (WFD) and Habitats Directive (HD) is lacking.

Recommendations to better include inland water ecosystems include the need to consider the following key issues:

- Specificity: Post-2020 targets should mention specifically the conservation of freshwater species and ecosystems, their genetic and functional diversity, and the linkages and dynamics between land and water.
- Justice: Given the integrated nature of freshwater ecosystems and the ecosystem services they provide that sustain human livelihoods, minimum requirements to achieve basic access to water should also be addressed.
- Targets and/or indicators: Amend, revise, or establish new targets or indicators, ensuring that they adequately represent the status of freshwater biodiversity and intimate linkage between land and water, including unreported interspecific and intraspecific biodiversity, habitat extent and condition, water quality and environmental flows, fisheries, presence of invasive species, extent and management of protected areas and other effective area-based conservation measure (OECMs), and delivery of ecosystem services.
- Monitoring: A more comprehensive monitoring of water quality and biodiversity is needed, including higher spatial and temporal resolution, consideration of "new" pollutants such as pesticides, microplastic, pharmaceutics, as well as mechanisms for adaptation to emerging ones. Monitoring of biodiversity should be comprehensive, addressing species diversity, as well as intraspecific diversity within and between populations.
- Coherence: Establish coherence and complementary among biodiversity directives and other directives such as the WFD. Several directives such as the WFD and HD generate extensive, spatially distributed datasets on aquatic biodiversity, however coherence and complementarity between these needs to be established and the data made available digitally, free of charge.

Link with EU Biodiversity strategy: Target 1—protect species and habitats; Target 2—maintain and restore ecosystems; Target 6—help stop the loss of global biodiversity.

## 4. Discussion

After some rapid discussion on the key messages (KMs) themselves, this section explores the role of MICESE process (cf. Figure 1) within existing science–policy processes, before outlining some limitations and added values of the MICESE approach.

### 4.1. Selected Comments on the Key Messages

To begin with, KM12 on freshwater biodiversity was much debated during the Ghent conference and within the editorial team. We finally decided to keep it, despite its relatively narrow focus and the suggestion received in Ghent conference to delete it. We kept it because we considered that freshwater ecosystems have been relatively overlooked in the previous biodiversity strategy despite having to face unique challenges. This does not mean that biodiversity associated with other types of ecosystems should not be considered, but rather that freshwater biodiversity deserves to be considered explicitly in the new Strategy.

Furthermore, no KM is specific to existing EU directives aimed at conserving biodiversity in Europe. This does not mean that these policies are no longer useful. They should be continued, but

with a stronger emphasis on effective implementation (e.g., of the HD) [15]. Our KMs highlight the global scale of the issue (KM1), which was already acknowledged by the current strategy—the need for a strategy of a systemic nature in terms of society choices, perceptions, and values [16,17], links with other sectors [15], and knowledge and research—and some key points that were not sufficiently acknowledged in the current strategy (large scales, climate change, biodiversity and ecosystem functions, freshwater biodiversity, etc.). On the specific point of KM6 (Knowledge and Research), the content of the KM and its place in the system of KMs both recognize the high value of research in this strategy as well as that different types of research and knowledge should be acknowledged [18–23].

We should also insist that our KMs contribute to the mainstreaming of existing analyses and ideas. Our KMs are not new, but stress issues already acknowledged by the scientific community but on which no clear solution or commitment has appeared so far. The challenge is now to find a way to deal with these issues in a political and efficient way.

As is usual in such exercises, we received comments that we were unable to include in our KMs since the process was finished. For instance, we received relevant comments from one of the reviewers of this paper, who suggested adding the following bullet points in KMs 6 and 7, respectively:

KM6—The greater participation of EU, governments, and/or national scientific centers in supporting (e.g., via dedicated system of grants) research on functioning of ecosystems and species in the face of biodiversity loss.

KM7—To make the EU member states more accountable for not implementing the Habitats Directive (HD) regulations; to simplify/accelerate procedures in the case of the destruction or disturbance of habitats especially those valuable in the context of climate change.

The rationale for the last proposition being, in the words of the reviewer, that: "Currently we can observe too often the tardy way of functioning of the European Commission in this aspect. Any infringement of the HD is dealt with by the Court of Justice of the European Union, which lengthens the procedure and habitats continue to be destroyed during this time. It would be useful to set up a body to deal directly with breaches of the HD and to be able to block negative actions immediately until the situation is clarified. In other words, galloping climate change does not allow us to waste time on overly complicated administrative procedures."

## 4.2. Comparison with Other Types of Multi-Phased Consultation Process

Science–policy interfaces are riddled with challenges, including mismatches between scientific and policy needs, diverse kinds of knowledge involved, inappropriate communication approaches, and potentially limited political will and scientific awareness [11,18–20,24–26]. In order to address some of these challenges, science–policy initiatives aim to improve the state of biodiversity both globally and in Europe. These include the work of the IPBES, EEA, and the European Platform for Biodiversity Research Strategy (EPBRS). The EEA recently published a report [27] on the state and outlook of the European environment containing a section on biodiversity and nature and KMs related to the topic. The KMs are mainly structured as findings on current status, but the report also contains recommendations for action relevant to the p2020EUBS. IPBES assessments on particular topics likewise often include knowledge gaps and policy recommendations. Finding ways to ensure that these and other existing efforts deliver scientific input for the p2020EUBS would definitely be beneficial. From 1999 to 2013, EPBRS was active through the support of two EU-funded projects (BioPlatform and BioStrat), developing research needs for biodiversity and ecosystem services. The approach of EPBRS consisted of developing research needs based on a topic selected by the member state holding the EU Presidency. Large-scale electronic conferences were organized on the topic to allow for a broad range of participants to suggest research priorities, which were then discussed and refined at EPBRS meetings where each member state (MS) sent one policymaker and one scientist who could contribute to the final joint development of research priorities, which were then disseminated at EU and MS level [28]. This approach, however, did not allow for an iterative dialogue around the recommendations. Instead, the process was participatory but rather linear in its approach.

From a methodological perspective, to compile research and evidence-based information on the most urgent EU issues on biodiversity and ecosystem services, different approaches could be used. EKLIPSE has gathered a wide range of knowledge synthesis methods [29], but it seems unlikely that any of them would have produced similar outputs and provided this type of multi-phased consultation and the opportunity to rank the relevance and robustness of the KMs and iterate them as thoroughly with different stakeholders. A more classical approach such as a literature review, possibly in combination with a questionnaire [30], or through a scientific ad-hoc group [31,32], could have been conducted to scan existing literature to discover what environmental issues are highlighted in current scientific literature, both quantitatively and qualitatively, and their urgency. Participatory research methods to synthesize expert input, such as a focus group [33] or Delphi process [34] could also have been used to gather input. Both methods allow the consultation of selected experts and their knowledge and experience through different processes generally from a limited size group. As we wished to remain as open and flexible (enabling any researcher to give input at any given time online) as possible in our consultation, focus groups or Delphis would have restricted these objectives. Lastly, we could have used horizon scanning at later phases of the consultation. Horizon scanning encompasses a range of (typically web-based) approaches for identifying emerging issues (or opportunities), such as innovations, associated impacts, risks, and benefits, by scanning the emerging literature (e.g., scientific, peer-reviewed, or otherwise) and then synthesizing this through knowledge management approaches [35].

The preparations for the p2020EUBS and the global framework strategy for biodiversity led by the CBD have attracted the attention of many organizations, scientific communities, EU MS [36], and most likely other efforts aiming to contribute elements that should be included in future strategies. However, we have yet to come across similar approaches to MICESE in terms of openly gathering messages from the broader scientific community. The process we outline in this paper reflects an iterative learning approach capable of dealing with many of the challenges of the science–policy interface, leading to a scientifically credible policy-relevant output. Whilst any of the above methods could have or could still be done to consolidate and complement our work, we argue that the MICESE method is itself a mix of methods deriving elements from some of the abovementioned traditional approaches. There were, however, limitations and challenges with the MICESE approach, and lessons learned for the application of the method in the future.

*4.3. Critical Reflection of the MICESE Method*

Similar to the horizon scanning exercise by [37,38], MICESE aimed to collect a list of suggestions and topics from the scientific community to feed into p2020EUBS. The response rate to the survey was not as high as we had hoped, despite it being circulated within large research institutions across Europe and on social media. This may be due to the novelty of the approach, the length of the survey as new KMs were added, or the length of the KMs themselves adding to the time required to carefully analyze the existing KMs and formulate new suggestions. This may also be explained by the high demands currently placed on researchers, who have less free time to attend such consultations: as an indicator of these demands, scientific journals are also struggling following high rates of scientists declining to carry out manuscript reviews.

In addition, the fact that the editorial teamwork in research organizations focused on environmental sciences may have had an impact on the dissemination of the call, resulting in fewer contributions from researchers based in universities. We also lacked contributions from researchers in the humanities and social science fields with only 10% of contributions coming from these researchers. This is an issue that we are confronted with in many science–policy initiatives but there are some valuable lessons we can learn from the MICESE process. The first is the need to word or frame the KMs so that they resonate as much as possible with researchers from both the natural and social sciences. The second is that to increase contributions from scientists, especially in the online consultation, much effort needs to be spent in communicating the purpose, timeline and added value of contributing to the wider scientific

community. For example, active presentation of the process to colleagues in Irstea (now INRAE) had a positive influence on their participation. Finally, a clear understanding of the target audience, and how to reach them and when, would also have helped to increase participation.

Another limitation at the beginning of our process was the limited opportunity for dialogue with the people suggesting issues or commenting on the KMs through the survey. Indeed, there were some comments that we felt we had to interpret, without being able to get back directly to survey participants to check our interpretation. Having a direct exchange with each participant would, however, have lengthened a process that was already fairly intensive. To a certain extent, this limitation was addressed at the Ghent conference, which offered greater opportunities for face-to-face interactions with contributors to the KMs.

Additionally, we initially planned to have a further online consultation on the KMs that emerged from Ghent conference, but did not perform it for two main reasons: first to respect the decisions and input from the participants at the Ghent conference; second, because we feared that an additional consultation after the review process would have delayed further the process of disseminating our final KMS to the European Commission. Finally, the end of the process—with the final KMs agreed on—is a little frustrating since we can no longer modify the KMs despite relevant suggestions (cf. end of Section 4.1).

Our decision to keep the process open to all individuals working in research institutions (and not, for example, only to people having a PhD) meant that there was no means to evaluate or check the credibility of the different inputs. Whilst our collegial, protocoled, and open way of incorporating inputs partly limited bias, we accept that there is no specific tool in MICESE to tackle this.

Despite the above limitations, the MICESE approach proved successful in a number of ways. The main added value of the approach we used here was the direct policy need—rather than the scientific need that often prevail in similar work—that drove the process of developing the KMs. In this sense, our process was very close to a user-push science–policy interface [18], the work having been commissioned to researchers by the EU Commission. The process allowed for the scientific community to suggest and comment on KMs to transfer to the European Commission and feed into the p2020EUBS. The process was not a closed shop, and encouraged all knowledge holders to participate, over a long period of time and using different means. The process also allowed policy makers from the European Commission to discuss with scientists the draft KMs at the Ghent conference. This allowed for a direct science–policy interface over a very targeted aim of the p2020EUBS. This meant that the KMs could be more closely aligned to the needs of decision-makers, and therefore increase the potential for uptake of the KMs in the strategy.

The mix of disciplinary background of the editorial team also allowed for an openness to the KMs proposed by the wider scientific community, and the ability to understand and address the comments received. For example, KMs were allocated to members of the editorial team with the expertise most relevant to the KM so that the processing of the KM would be the most adapted. This may be central since solving environmental issues is not only reliant on environmental sciences, but also broader cooperation and breaking silos between different disciplines [16,19], as acknowledged by some of our KMs (e.g., KM3 and KM4). Not only was our group made up of researchers from different disciplines, it was also composed of researchers with experience in interdisciplinary work and in science–policy or science–management interfaces. This helped greatly in the team developing KMs that were user-oriented. A final key ingredient was that many of us in the editorial team had known and worked with each other previously, resulting in a team coherence and cohesion, which made the editorial process much more streamlined and effective. This specificity, however, had the drawback of a possibly increased risk of group thinking.

The conference also gave researchers and other participants the opportunity to reflect on the scientific KMs against the work of the societal KM process in a workshop, which helped illustrate the diverse ways that different actors conceive biodiversity issues as important and inspire dialogue on them from a new perspective. While the societal KMs were more pragmatic and focused on the

individual's everyday actions, the underlying concerns and casualties were similar to those expressed in the scientific KMs. Showing how the scientific KMs can aid to solve the concerns voiced in the societal KMs is important for increased impact of both messages and learning about science–policy–society communication and collaboration [12].

The multiphased nature of MICESE was also a strength. Having an internet survey before discussions during the conference allowed to open up the variety and breadth of participants contributing to the process.

We argue that MICESE, leading to the development of the KMs, could and should be used in other processes that aim to respond to a policy need by coalescing the views of the scientific community. The process allowed for limited bias from the editing team in the selection and drafting of KMs and led to an output adapted to the needs of decision-makers. In addition, the process of developing the KMs was open and transparent, leading to scrutiny of the process at every stage.

## 5. Conclusions

Through the development of the p2020EUBS, the scientific community of the ALTER-Net and EKLIPSE networks saw a unique opportunity to make a difference for biodiversity in Europe by better involving scientists, policy makers, and society. This paper reports the KMs from the scientific community for the p2020EUBS. Our novel and adaptive MICESE procedure of KM building allowed us to highlight 12 KMs, mostly mainstreaming already existing elements into a coherent set of KMs for the European Commission to use in preparing the new EU Biodiversity Strategy. The conditions that made such a process based on MICESE successful were the presence of a policy request that drove the overall process, an interdisciplinary editorial team, and a range of processes to increase involvement of the wider scientific community. The policy need for the development of the KMs came from the European Commission to the ALTER-Net scientific network. There was therefore a direct mandate from the organization responsible for developing the p2020EUBS to coalesce the views of the scientific community and feed this into the development of the strategy. In addition, the policy requester was included in the development of the KMs, through the invitation to see, comment on, and discuss the KMs at the Ghent conference, together with the scientific community. This allowed not only for the requester to be kept informed of the direction and composition of the KMs, but also allowed researchers to better understand the needs of policymakers through discussion of the KMs. The interdisciplinary nature of the editorial team was a key condition leading to the success of this approach. This allowed for the flexibility of allocating the KMs to individuals knowledgeable about the themes covered in the KMs and prevented any KMs from being neglected or badly managed. Finally, through the open, transparent, and varied nature of engagement process, we enabled researchers to feed into the editing process at different times and through different means. The workshop in Peyresq, online surveys and workshops, posters, and plenaries in Ghent all enabled researchers to choose how to become engaged. We would strongly suggest using such a range of formats in future processes to ensure that scientists can become involved in ways tailored to their needs and preferences.

We expect our work on the identification of key messages will have an impact in the new EU biodiversity strategy and we are looking forward to the incorporation of our work in larger coalitions. As a first step in this direction, we are planning a reflection around both our scientific and societal KMs [12]) within the ALTER-Net and EKLIPSE scientific networks to devise how they should be communicated, especially in terms of the links between the societal and scientific inputs. Following on from this reflection, we will explore ways in which MICESE could be used in other processes that aim to respond to a policy need by coalescing the views of the scientific community.

**Supplementary Materials:** The following are available online at http://www.mdpi.com/2071-1050/12/6/2385/s1, Supplementary File S1: example of successive corrections for one of the Key Messages (KM originally numbered 1, corresponding to final KM7); Supplementary File S2: editorial team rules; Supplementary File S3: correspondence between initial numbers and versions of key messages and final ones.

**Author Contributions:** Conceptualization, all authors; methodology, all authors; validation, all authors; investigation, all authors; resources, all authors; data curation, all authors; writing—original draft preparation, all authors; writing—review, all authors; editing, J.Y., A.W., and all other authors; visualization, L.V. and F.G.; supervision, F.G.; project administration, F.G.; funding acquisition, M.V. All authors have read and agreed to the published version of the manuscript.

**Funding:** This article arose from an EKLIPSE project activity (EKLIPSE grant agreement number 690474, European Union's Horizon 2020 research and innovation programme).

**Acknowledgments:** We warmly thank Maurice Hoffmann (INBO) and ALTER-Net for initiating the request that led to the process, Anne-Gerdien Prins (PBL) for her involvement in the beginning of the scientific task force process, Thibault Datry (INRAE Lyon) for his participation at the beginning of the editorial team before his H2020 project did not allow him to attend our meetings, other participants to the Peyresq initial meeting: Gregor Kalinkat (IGB Berlin), Amy McDougall (JNCC), Carlos Romao (EEA), Janine van Vessem (INBO), Marie Dolores Lopez Rodriguez (CAESCG), Philip Roche (INRAE Aix-en-Provence), Martin Sharman, Stefano Targetti (INRAE Avignon), the online survey participants and the participants to ALTER-Net & EKLIPSE Ghent conference. We also thank three reviewers for their useful insights on a previous version of this manuscript.

**Conflicts of Interest:** The authors declare no conflict of interest. The funders had no role in the design of the study; in the collection, analyses, or interpretation of data; in the writing of the manuscript, or in the decision to publish the results.

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
