# Peer review of "MICESE: A New Method Used for the Formulation of Key Messages from the Scientific Community for the EU Post 2020 Biodiversity Strategy"

_sustainability, doi:10.3390/su12062385_

Round 1

Reviewer 1 Report

This paper reports on the process of developing Key Messages for the post-2020 EU biodiversity strategy, as developed and run by the ALTERNET and EKLIPSE consortia. The findings are 12 key messages which have been developed through an interesting iterative consultation and reviewing process.

Overall, I feel that this is a very important paper which will be widely used, with important outcomes but also reporting on a lengthy and impressive consultation process helping to build greater confidence in those outcomes.

I have made some suggestions for small edits and points of clarification in the MS file directly (and hopefully I have managed to attach this to these notes!). In addition there are a few other points that I think the authors need to consider:

l. 96 – There is mention of a societal process but we don’t find out more about it. Is it important and does it impact on the outcome of the science process? If so, maybe we need to know more about it…?

l. 168-170. Is there any way the early iterations of KMs could be made available, even as an online appendix? This would make it possible for anyone to follow this iterative process through time. Even if it was just a headline sentence saying which ones were merged to create new ones at each stage. Or is this information already available elsewhere? If so, maybe this should be clear?

I have given less detailed commentary on the KMs themselves – they come from of a lengthy and detailed reviewing process and so I think they should stand as the outputs of that process and I don’t feel it’s my job to tinker with them. However, I have read them from a sense-check perspective and have made a couple of suggestions for improving clarity that the authors might like to consider.

Reflection on the process - Is there any risk in not taking KMs back to the wider scientific community after the internal editing process? I appreciate there will always be disagreement from scientists, so you’ll never get consensus, but is there a risk of group think?

Author Response

This paper reports on the process of developing Key Messages for the post-2020 EU biodiversity strategy, as developed and run by the ALTERNET and EKLIPSE consortia. The findings are 12 key messages which have been developed through an interesting iterative consultation and reviewing process.

Overall, I feel that this is a very important paper which will be widely used, with important outcomes but also reporting on a lengthy and impressive consultation process helping to build greater confidence in those outcomes.

I have made some suggestions for small edits and points of clarification in the MS file directly (and hopefully I have managed to attach this to these notes!).

We thank you for this detailed re-reading, which was very useful. We indeed received the pdf file and agreed with most of the suggestions made. Below are the only points we only partially took into account, with justifications for each:

- Table 2 (now Table 1): the wording of the possible answers was not changed because it was the wording used for the consultation. Also, we do not give the results for the “no” answers because they had less meaning in case of fused messages.

- Figure 2: two remarks/ answers here:

  • There is no clear left/right axis here. We could have devised one for the first row of key messages but it would not have applied to the other rows.
  • We have kept the reference to driver, state and response but without a reference to the DPSIR framework. For us, KM10 is clearly not a driver but rather a response of society to the biodiversity erosion problem.

- Line 460: change “still Judged” to “also to be”: we did not see the problem and kept “still judged”

 In addition there are a few other points that I think the authors need to consider:

  1. 96 – There is mention of a societal process but we don’t find out more about it. Is it important and does it impact on the outcome of the science process? If so, maybe we need to know more about it…?

The societal process is described in more detail in a different paper in this same special issue (Varumo et al 2020) which has now been referred to in the text lines 109-110 (version with visible modifications) (Material and methods) and towards the end of the discussion and in the conclusion. It was a parallel process that did not influence the content of the scientific key messages, but rather gave an opportunity to reflect different ways to express environmental concerns and future needs and the synergies are analysed in more detail in the Varumo et al paper.

  1. 168-170. Is there any way the early iterations of KMs could be made available, even as an online appendix? This would make it possible for anyone to follow this iterative process through time. Even if it was just a headline sentence saying which ones were merged to create new ones at each stage. Or is this information already available elsewhere? If so, maybe this should be clear?

This is a very good remark. We indeed hesitated to make it available extensively. We propose a middle ground: putting as supplementary material the list of key message versions prior to the final internal review and showing the files we used for one KM that was not fused.

I have given less detailed commentary on the KMs themselves – they come from of a lengthy and detailed reviewing process and so I think they should stand as the outputs of that process and I don’t feel it’s my job to tinker with them. However, I have read them from a sense-check perspective and have made a couple of suggestions for improving clarity that the authors might like to consider.

Thank you for this. You interpreted the status of these KMs perfectly. We did take into account the changes you proposed.

Reflection on the process - Is there any risk in not taking KMs back to the wider scientific community after the internal editing process? I appreciate there will always be disagreement from scientists, so you’ll never get consensus, but is there a risk of group think?

This is a very good question. It was indeed originally planned but we decided against it in part to avoid diluting the input from the participants of the Ghent conference, but also to make the most of the policy window in terms of the key messages. We have added some points to this effect in 780-787.

Reviewer 2 Report

Highlight changes in yellow in a next revision, please. No track changes.

Consider comments in the entire text.

Although “EU” is s known abbreviation, consider defining it

Upper and lower latter presented in names, check template and coherence, as in affiliations starting by abbreviations, non-defined…

Should an abstract start like this: “Abstract: The EU 2020 Biodiversity strategy will soon come to an end and may not have been as 23 successful as envisioned.”

See there is no previous contextualization, to be followed by methodology, findings and practical implications, revise structure…

Define all abbreviations, I know what EKLIPSE is, many do not…

Do not use “We”

Consider entirely revising language according to suggested structure…

“ We provide insights and analyses of the 33 new method through which these key messages were generated and processed. We finally reflect 34 on how to improve the future involvement of scientists in science-policy interfaces.”

Avoid repeating sentences through the text…

“ The EU 2020 Biodiversity strategy will soon come to an end”

And this time with a reference…

1. Introduction 39

The current EU Biodiversity Strategy to 2020 will soon come to an end [1].”

All abbreviations must be revised, and first defined… “IPBES”

Authors should try to avoid this kind of colloquial language in a scientific article… “Despite, and perhaps because of, the failure to meet the 2010 target and now the pessimistic 54 prognosis in meeting the 2020 target, biodiversity loss remains a major societal concern.”

Again, revise language to remove “We” and be assertive: “In this paper, we describe the processes we used to gather input from the scientific community 77 to elaborate what we call scientific key messages for the post 2020 EU biodiversity strategy. We call 78 this method the Multiphased, iterative and consultative elicitation of scientific expertise (MICESE) 79 method. By reflecting on this method, we analyse what type of messages a multiphased consultation 80 process produced and how they developed, we identify the advantages and disadvantages of the 81 way in which we worked compared to other existing approaches. Finally, and finally we consider the 82 lessons learnt for improved involvement of researchers in science-policy interfaces.”

At this stage, only general purpose should be addressed, consider moving other conte to abstract, discussion, conclusions, implications or limitations…

In my perspective, the use of footnotes should be avoided in order to have a clearer manuscript, avoiding the type of writing used in  report. This is a scientific article.

“1 Europe's ecosystem research network, see http://www.alter-net.info

2 Establishing a European Knowledge and Learning Mechanism to Improve the Policy-Science-Society Interface on Biodiversity and Ecosystem Services, project funded by European Union’s Horizon 2020 Programme for research and innovation. See http://www.eklipse-mechanism.eu/”

Do not use abbreviations alone in tables that may be consulted separately by readers, add notes as the end, then…

Make reading easier…

A caption must be self-explanatory and clear so that the reader may move forward and quickly understand what is being presented…

Not like this, move content to text, then…

Table 3. Outputs of the online consultation phase on the policy relevance and scientific basis of the 206 key messages. In each category, we kept the maximum score in case key messages were merged. In 207 terms of policy relevance, the question was: “I agree this key message is important to take into account 208 in EU biodiversity post 2020”, with a score between 0 and 10. Regarding the scientific basis, the 209 participants to the survey were asked about the “level of scientific knowledge substantiating the key 210 message” with 6 possible levels: “Well established”, “Partially established”, “Mixed (knowledge pro 211 and con”, “contra (knowledge mostly against the key message)”, “Unknown scientifically”, “I don’t 212 know”. We here give the frequency of “Well established” answers.”

Figure 2 has no quality and is difficult to understand, no connection between KMs

Do not define abbreviations more than one a tet like this will need to be deeply checked before submission…

“Biodiversity and Ecosystem 229 Services (IPBES). ”

“and the work of the Intergovernmental Science-Policy Platform on Biodiversity and Ecosystem 235 Services (IPBES). ”

Check footnotes information….

“3 https://ec.europa.eu/research/environment/index.cfm?pg=nbs

4 https://www.hutton.ac.uk/sites/default/files/files/research/srp2016-21/MEEM%20Technical%20Report%20(Nov%202017).pdf  ”

Such extensive lists during the text do show this is a report, than need to be translated in a scientific text…

Well, it seems to me that discussion should integrate section 3. I fact it is easy to “list” and then to discuss separately.

Understanding the intention, I have difficulty in seeing what is really relevant here…

See that this could be at the end at conclusions or in a separate limitations section, not here…

“Despite the above limitations, the MICESE approach proved successful in a number of ways. 651 The main added value of the approach we used here was the direct policy need - rather than the 652 scientific need that often prevail in similar work - that drove the process of developing the key 653 messages.”

Conclusions should follow a somewhat similar structure to abstract, mentioned above.

Context

Methods

Results

Implications

Again, I need findings to be mentioned in abstract, so that looking at title, abstract and conclusions, I feel the need to read the entire text.

I am very aware of the aims and process of EKLIPSE. I do feel authors must translate what was done and achieved in amore enlightening way, avoiding the report form…

This developed work is he and it involved an iterative process and many collaborations.

This text intends to describe a very long process, but readers do expect to see clear findings

This is more a report, than a scientific comprehensive text

Authors do need to further work the text to make it more relevant and enlightening, accessible to the entire scientific community in the environmental area, dealing with collection of information and decision methods

References are somewhat scarce, I must say…

Consider all comments, in presented supplementary material.

Deeper statistics would be important…

Author Response

 Consider comments in the entire text.

Although “EU” is s known abbreviation, consider defining it

Thank you. We have defined it now at the beginning of the introduction.

Upper and lower latter presented in names, check template and coherence, as in affiliations starting by abbreviations, non-defined…

We have changed the case of the name of the first authors (it was a mistake). Regarding the affiliations, we have not changed them since these formats are imposed by some of our institutes.

Should an abstract start like this: “Abstract: The EU 2020 Biodiversity strategy will soon come to an end and may not have been as 23 successful as envisioned.”

We see the imminent end of the current EU biodiversity strategy as such an important driver for the work reported here that we would like to retain it.

See there is no previous contextualization, to be followed by methodology, findings and practical implications, revise structure…

We have made changes to the abstract - especially in terms of the findings (also in response to another remark), having in mind the structure required by the journal. We hope the new version fits with your suggestion.

Define all abbreviations, I know what EKLIPSE is, many do not…

EKLIPSE is now defined in the introduction, as well as ALTER-Net. If the editor prefers that we also define them in the abstract we are ready to do so as well.

Do not use “We”

We considered this suggestion but as the MICESE process was a very dynamic and active process, we felt this was better reflected with the use of “we”.

Consider entirely revising language according to suggested structure…

“ We provide insights and analyses of the 33 new method through which these key messages were generated and processed. We finally reflect 34 on how to improve the future involvement of scientists in science-policy interfaces.”

 Unfortunately we are unsure what the reviewer is suggesting as the above sentence seems to be a repetition of the two last sentences of our abstract.

Avoid repeating sentences through the text…

“ The EU 2020 Biodiversity strategy will soon come to an end”

And this time with a reference…

1. Introduction 39

The current EU Biodiversity Strategy to 2020 will soon come to an end [1].”

These two sentences are indeed in the abstract and in the text itself. According to our experience, it is common practice in research that some sentences or parts of sentences are repeated in the abstract and the main text.

All abbreviations must be revised, and first defined… “IPBES”

We agree that we were not thorough enough in the treatment of abbreviations within the text (KM, ALTER Net, EEA…). This has now been changed. And we agree with the remark on IPBES.

Authors should try to avoid this kind of colloquial language in a scientific article… “Despite, and perhaps because of, the failure to meet the 2010 target and now the pessimistic 54 prognosis in meeting the 2020 target, biodiversity loss remains a major societal concern.”

We have removed “, and perhaps because of,”

Again, revise language to remove “We” and be assertive: “In this paper, we describe the processes we used to gather input from the scientific community 77 to elaborate what we call scientific key messages for the post 2020 EU biodiversity strategy. We call 78 this method the Multiphased, iterative and consultative elicitation of scientific expertise (MICESE) 79 method. By reflecting on this method, we analyse what type of messages a multiphased consultation 80 process produced and how they developed, we identify the advantages and disadvantages of the 81 way in which we worked compared to other existing approaches. Finally, and finally we consider the 82 lessons learnt for improved involvement of researchers in science-policy interfaces.”

If we were to replace “we”, we would be less assertive (and readable). We have therefore kept this sentence.

At this stage, only general purpose should be addressed, consider moving other conte to abstract, discussion, conclusions, implications or limitations…

We do not know which part of the manuscript this refers to so we are afraid we cannot take it into account. We also do not know what “conte” refers to.

In my perspective, the use of footnotes should be avoided in order to have a clearer manuscript, avoiding the type of writing used in  report. This is a scientific article.

“1 Europe's ecosystem research network, see http://www.alter-net.info

2 Establishing a European Knowledge and Learning Mechanism to Improve the Policy-Science-Society Interface on Biodiversity and Ecosystem Services, project funded by European Union’s Horizon 2020 Programme for research and innovation. See http://www.eklipse-mechanism.eu/”

The guidelines for authors of Sustainability provide no guidance on footnotes but if the Editor requests there removal we will provide the information contained in them in a different way.

Do not use abbreviations alone in tables that may be consulted separately by readers, add notes as the end, then…

Make reading easier…

We agree with the remark on tables. We have therefore redefined some abbreviations within legends of tables or figures.

A caption must be self-explanatory and clear so that the reader may move forward and quickly understand what is being presented…

Not like this, move content to text, then…

Table 3. Outputs of the online consultation phase on the policy relevance and scientific basis of the 206 key messages. In each category, we kept the maximum score in case key messages were merged. In 207 terms of policy relevance, the question was: “I agree this key message is important to take into account 208 in EU biodiversity post 2020”, with a score between 0 and 10. Regarding the scientific basis, the 209 participants to the survey were asked about the “level of scientific knowledge substantiating the key 210 message” with 6 possible levels: “Well established”, “Partially established”, “Mixed (knowledge pro 211 and con”, “contra (knowledge mostly against the key message)”, “Unknown scientifically”, “I don’t 212 know”. We here give the frequency of “Well established” answers.”

We have changed the legend for clarity, but kept the content useful to understand the table independently of the text. If we had moved the content to the text, the Table would have no longer been readable alone.

Figure 2 has no quality and is difficult to understand, no connection between KMs

We think Figure 2 is useful to represent the structure of the 12 KMs on two axes (systemic value, scale and driver/state/response status) but we have changed Figure 2 together with its legend based on constructive comments we received on it. Putting connections between KMs would have made the Figure too complex. If the Editor considers the Figure to be of low quality we will change it.

Do not define abbreviations more than one a tet like this will need to be deeply checked before submission…

“Biodiversity and Ecosystem 229 Services (IPBES). ”

“and the work of the Intergovernmental Science-Policy Platform on Biodiversity and Ecosystem 235 Services (IPBES). ”

We agreed (and changed) except in the case of tables, figures (cf. above).

Check footnotes information….

“3 https://ec.europa.eu/research/environment/index.cfm?pg=nbs

4 https://www.hutton.ac.uk/sites/default/files/files/research/srp2016-21/MEEM%20Technical%20Report%20(Nov%202017).pdf  ”

We checked them and they are correct.

Such extensive lists during the text do show this is a report, than need to be translated in a scientific text…

Maybe the reviewer is referring to the key messages themselves? We hesitated to put them in supplementary material, but as they are the main results of our work we have decided to keep them in the results section. This can be discussed with the Editor in case of disagreement but the other reviewers saw the value of keeping the Key Messages in the main text and we believe the paper would lose most of its impact if they were moved to supplementary material.

Well, it seems to me that discussion should integrate section 3. I fact it is easy to “list” and then to discuss separately.

Some of the points in the discussion do not relate to the results section but to the M&M section. As such, we would argue that  having a separate discussion is a better alternative for our paper.

Understanding the intention, I have difficulty in seeing what is really relevant here…

We are sorry but we do not understand what this remark refers to in our paper.

See that this could be at the end at conclusions or in a separate limitations section, not here…

“Despite the above limitations, the MICESE approach proved successful in a number of ways. 651 The main added value of the approach we used here was the direct policy need - rather than the 652 scientific need that often prevail in similar work - that drove the process of developing the key 653 messages.”

Many thanks for your suggestion. We have kept this text where it is but we have also added some of this commentary in the conclusions sections as well (see line 834 (version with visible modifications)).

Conclusions should follow a somewhat similar structure to abstract, mentioned above.

Context

Methods

Results

Implications

We have no experience of such a structure for the conclusion section and this is not referred to in the Sustainability guidelines. We have therefore kept the structure of the conclusion as it is.

Again, I need findings to be mentioned in abstract, so that looking at title, abstract and conclusions, I feel the need to read the entire text.

We agree on this remark and have added more information on findings in the abstract.

I am very aware of the aims and process of EKLIPSE. I do feel authors must translate what was done and achieved in amore enlightening way, avoiding the report form…

This developed work is he and it involved an iterative process and many collaborations.

This text intends to describe a very long process, but readers do expect to see clear findings

This is more a report, than a scientific comprehensive text

Authors do need to further work the text to make it more relevant and enlightening, accessible to the entire scientific community in the environmental area, dealing with collection of information and decision methods

We have reworked the text based on the comments received. To address the point that our text was of a report nature, we have extended the discussion and included further relevant references.

References are somewhat scarce, I must say…

We agree with this remark. We have added references to meet standards in Sustainability and other scientific journals.

Consider all comments, in presented supplementary material.

The remark is unclear.

Deeper statistics would be important…

We do not see which kind of deeper statistics would be useful for the aim and content of the paper but would welcome further suggestions.

Reviewer 3 Report

The paper of Gosselin et al. entitled: “Key messages from the scientific community for the EU post 2020 Biodiversity Strategy - elaboration of the Multiphased, Iterative and Consultative Elicitation of Scientific Expertise (MICESE) method” rises very important and topical issue relating to the challenges facing the new strategy for the protection of biodiversity in Europe after 2020. The Authors present in the paper 12 key messages for the new strategy for biodiversity which were selected and formulated using method called MICESE. This method combined the scientific knowledge and policy-makers experience and was refined over a series of meetings, conferences and workshops. Compared to other methods used so far in participatory research it is distinguished by its openness, flexibility as well as multilevel approach. I am convinced that this paper will be or should be the object of interest of wide scientific audience. I can only hope that the recommendations made by Authors (and their respondents) will be applied in the nearest future.

The paper meets the requirements of the Sustainability journal and can be published after including the below mentioned comments.

Major comments:

Title:

In my opinion it is too long. There is no need to present the full name of the method here. It can be mentioned in the abstract for the first time. Then you will avoid the problem with the keywords – see below.

New title for consideration: MICESE – a new method of formulation of the key messages for the EU post 2020 Biodiversity Strategy

Abstract:

Well-written and informative. However, if you keep the title as it is, there is no need to develop here MICESE abbreviation.

Keywords:

We usually avoid repeating the same words in title and keywords – biodiversity, strategy, elicitation of scientific expertise; iterativity. According to the Sustainability Journal requirements, I would suggest changing these duplicated keywords.

Methods:

The method part is rich in details which refer to the places and dates of conferences and meetings. It makes all this part a little bit convoluted. I believe it is possible to present some part of this information in the form of table and make this fragment of the manuscript more concise. The terms “policy relevance scores” and “scores” are mentioned for the first time in the results in the main text as well as in the description of Table 3. However, a brief explanation of the process how they were counted, based on what questions in the online survey should be included in the material and methods part.

Results:

The presented KMs are generally described and justify sufficiently. It was difficult to add something more. The strong side of this paper is also that the Authors propose new targets for the post-2020 EU biodiversity strategy (e.g. KMs 3, 4, 9, 11) or enhance the role of other targets especially the need for protecting inland water biodiversity (KM12). Below you will find some suggestions for additional recommendations for KMs 6 and 7. Feel free to use them if you like.

KM6 – The greater participation of EU, governments and/or national scientific centres in supporting (e.g. via dedicated system of grants) research on functioning of ecosystems and species in the face of biodiversity loss.

KM7 – To make the EU member states more accountable for not implementing the Habitats Directive (HD) regulations; to simplify/accelerate procedures in the case of the destruction or disturbance of habitats especially those valuable in the context of climate change.

Currently we can observe too often the tardy way of functioning of the EC in this aspect. Any infringement of the HD is dealt with by the CJEU, which lengthens the procedure and habitats continue to be destroyed during this time. It would be useful to set up a body to deal directly with breaches of the HD and to be able to block negative actions immediately until the situation is clarified. In other words, galloping climate change does not allow us to waste time on overly complicated administrative procedures.

Discussion

I found the discussion very interesting. I appreciate the Authors underline the pros and cons of the MICESE method simultaneously. There is only the lack of the reference to the latest paper of Sutherland et al. 2019. A Horizon Scan of Emerging Global Biological Conservation Issues for 2020.

Minor comments:

Line 44: Please develop EEA abbreviation. Abbreviations should be defined in parentheses the first time they appear in the abstract, main text, and in figure or table captions and used consistently thereafter.

Line 47: Please develop IPBES abbreviation.

Line 52: recognise instead of recognises

Line 71: EU post 2020 Biodiversity Strategy - the statement [hereafter called p2020EUBS] should be in this place

Line 82: delete “and finally”

Line 84: Leave only “Material and methods” nothing more

Lines 95-96: “input to the post-2020 EU biodiversity strategy [hereafter called p2020EUBS]” should be changed into “input to the p2020EUBS”

Lines 104-105 and 710: AlterNet and Eklipse should be in capital letters

Lines 127-140: If you decide to use abbreviation KM for key message, please use it consequently throughout the manuscript within all its parts.

Lines 162-163: Do not develop the MICESE abbreviation.

Line 180: typos in the word input

Lines 177-183: In my opinion the Table 1 is redundant. You should put all the information in the text and do not repeat it in the table.

Lines 229-230 and 235-236: Leave only IPBES abbreviation.

Lines 231-236: This point is an exact copy of the first point.

Line 271: well-being instead of wellbeing

Line 377: Leave only the abbreviation

Line 531: Think about using HD and WFD abbreviations also earlier in the text. You mentioned these legislations very often and it should be unified throughout the text.

Lines 536-539: You partially repeat here what has been written earlier in lines 530-531. I suggest to join this fragment with the subsection referring to coherence – then you can reverse the order: specificity, justice, targets, monitoring, coherence.

Line 543-545: I suggest removing this paragraph.

Lines 547-548: I suggest removing: “Before discussing the MICESE approach, some short comments on the KMs themselves are useful to complement the KMs above:”

Lines 549, 555, 563: First, Second, Third should be changed into Firstly, Secondly, Thirdly

Line 573: Leave only the abbreviations.

Lines 607, 621: remove: “post-2020 EU biodiversity strategy”

Line 628: lack of spacebar in sciencemay

Line 670: comma between 7 and 9

Conclusions: Use KMs abbreviation were needed.

In the results, when you describe key messages and give recommendations one time you start the subsections from the capital letter, the other time do not. Please, unify this throughout this part.

Author Response

The paper of Gosselin et al. entitled: “Key messages from the scientific community for the EU post 2020 Biodiversity Strategy - elaboration of the Multiphased, Iterative and Consultative Elicitation of Scientific Expertise (MICESE) method” rises very important and topical issue relating to the challenges facing the new strategy for the protection of biodiversity in Europe after 2020. The Authors present in the paper 12 key messages for the new strategy for biodiversity which were selected and formulated using method called MICESE. This method combined the scientific knowledge and policy-makers experience and was refined over a series of meetings, conferences and workshops. Compared to other methods used so far in participatory research it is distinguished by its openness, flexibility as well as multilevel approach. I am convinced that this paper will be or should be the object of interest of wide scientific audience. I can only hope that the recommendations made by Authors (and their respondents) will be applied in the nearest future.

The paper meets the requirements of the Sustainability journal and can be published after including the below mentioned comments.

We thank you very much for these positive comments.

Major comments:

Title:

In my opinion it is too long. There is no need to present the full name of the method here. It can be mentioned in the abstract for the first time. Then you will avoid the problem with the keywords – see below.

New title for consideration: MICESE – a new method of formulation of the key messages for the EU post 2020 Biodiversity Strategy

Abstract:

Well-written and informative. However, if you keep the title as it is, there is no need to develop here MICESE abbreviation.

We warmly thank the reviewer for this suggestion. We agree that the title was too long. We propose another one based on your suggestion: “MICESE: a new method used for the formulation of key messages from the scientific community for the EU post 2020 Biodiversity Strategy”. Given this, there is a need to develop MICESE in the abstract.

Note that based on remarks by other reviewers we have somewhat changed the abstract.

Keywords:

We usually avoid repeating the same words in title and keywords – biodiversity, strategy, elicitation of scientific expertise; iterativity. According to the Sustainability Journal requirements, I would suggest changing these duplicated keywords.

Based on this comment and the new title, we removed biodiversity and strategy from the title. We have however not found in the instructions for authors that keywords should not repeat title words. If the Editor would like us to do so, we will reintroduce these keywords.

Methods:

The method part is rich in details which refer to the places and dates of conferences and meetings. It makes all this part a little bit convoluted. I believe it is possible to present some part of this information in the form of table and make this fragment of the manuscript more concise.

We thank the reviewer for this comment and have made efforts to shorten the section to two pages. We have attempted to add sufficient detail for readers to be able to replicate the method. To make it visually more appealing, we have synthesized the information in Figure 1.  

The terms “policy relevance scores” and “scores” are mentioned for the first time in the results in the main text as well as in the description of Table 3. However, a brief explanation of the process how they were counted, based on what questions in the online survey should be included in the material and methods part.

The reviewer is absolutely right. This was partly mentioned in the M&M, but too implicitly. We have therefore added some elements in the M&M.

Results:

The presented KMs are generally described and justify sufficiently. It was difficult to add something more. The strong side of this paper is also that the Authors propose new targets for the post-2020 EU biodiversity strategy (e.g. KMs 3, 4, 9, 11) or enhance the role of other targets especially the need for protecting inland water biodiversity (KM12). Below you will find some suggestions for additional recommendations for KMs 6 and 7. Feel free to use them if you like.

KM6 – The greater participation of EU, governments and/or national scientific centres in supporting (e.g. via dedicated system of grants) research on functioning of ecosystems and species in the face of biodiversity loss.

KM7 – To make the EU member states more accountable for not implementing the Habitats Directive (HD) regulations; to simplify/accelerate procedures in the case of the destruction or disturbance of habitats especially those valuable in the context of climate change.

Currently we can observe too often the tardy way of functioning of the EC in this aspect. Any infringement of the HD is dealt with by the CJEU, which lengthens the procedure and habitats continue to be destroyed during this time. It would be useful to set up a body to deal directly with breaches of the HD and to be able to block negative actions immediately until the situation is clarified. In other words, galloping climate change does not allow us to waste time on overly complicated administrative procedures.

We thank you very much for your appreciation of our work as well as the two added comments - which are indeed relevant. As the process has come to an end, we have added the comments and our appreciation specifically at the end of section 4.1.

Discussion

I found the discussion very interesting. I appreciate the Authors underline the pros and cons of the MICESE method simultaneously. There is only the lack of the reference to the latest paper of Sutherland et al. 2019. A Horizon Scan of Emerging Global Biological Conservation Issues for 2020.

We thank you very much for this positive comment and for the reference. We have added it.

Minor comments:

Line 44: Please develop EEA abbreviation. Abbreviations should be defined in parentheses the first time they appear in the abstract, main text, and in figure or table captions and used consistently thereafter.

 and:

Line 47: Please develop IPBES abbreviation.

Thank you. We agree and have done it.

Line 52: recognise instead of recognises

Thank you. Done.

Line 71: EU post 2020 Biodiversity Strategy - the statement [hereafter called p2020EUBS] should be in this place

Thank you. Done. We have also replaced the entire expression by its acronym in some places in the text.

Line 82: delete “and finally”

Thank you. Done.

Line 84: Leave only “Material and methods” nothing more

Thank you. Done.

Lines 95-96: “input to the post-2020 EU biodiversity strategy [hereafter called p2020EUBS]” should be changed into “input to the p2020EUBS”

 Thank you. Done.

Lines 104-105 and 710: AlterNet and Eklipse should be in capital letters

 Thank you. Done. Note that ALTER-Net is written in a mixture of upper and lower case.

Lines 127-140: If you decide to use abbreviation KM for key message, please use it consequently throughout the manuscript within all its parts.

Thank you. A priori done.

Lines 162-163: Do not develop the MICESE abbreviation.

We hesitate on this one. As the acronym is only explained in the abstract, we prefer to keep the complete name here. In case the Editor wishes the reverse, we will change this.

Line 180: typos in the word input

Thank you. Corrected.

Lines 177-183: In my opinion the Table 1 is redundant. You should put all the information in the text and do not repeat it in the table.

Thank you for the relevant suggestion. We have done as suggested.

Lines 229-230 and 235-236: Leave only IPBES abbreviation.

Thank you. Done.

Lines 231-236: This point is an exact copy of the first point.

This is correct. It has been corrected.

Line 271: well-being instead of wellbeing

Thank you. Corrected.

Line 377: Leave only the abbreviation

Thank you. Corrected.

Line 531: Think about using HD and WFD abbreviations also earlier in the text. You mentioned these legislations very often and it should be unified throughout the text.

Thank you. Done.

Lines 536-539: You partially repeat here what has been written earlier in lines 530-531. I suggest to join this fragment with the subsection referring to coherence – then you can reverse the order: specificity, justice, targets, monitoring, coherence.

Very good suggestion. Thank you. Since it does not change the meaning of the KM, just its presentation, we have performed the suggestion.

Line 543-545: I suggest removing this paragraph.

 Lines 547-548: I suggest removing: “Before discussing the MICESE approach, some short comments on the KMs themselves are useful to complement the KMs above:”

We have actually kept the first paragraph (former lines 543-545) and removed the second (547-548), that was simply repeating what had been said in the former.

Lines 549, 555, 563: First, Second, Third should be changed into Firstly, Secondly, Thirdly

Thank you. Changed as suggested.

Line 573: Leave only the abbreviations.

Thank you. Done.

Lines 607, 621: remove: “post-2020 EU biodiversity strategy”

Thank you. Done.

Line 628: lack of spacebar in sciencemay

Thank you. Corrected.

Line 670: comma between 7 and 9

Thank you. Corrected.

Conclusions: Use KMs abbreviation were needed.

Thank you. Done.

In the results, when you describe key messages and give recommendations one time you start the subsections from the capital letter, the other time do not. Please, unify this throughout this part.

Thank you. Done.

Round 2

Reviewer 2 Report

Highlight changes in yellow in a next revision, please. No track changes.

Consider comments in the entire text.

The way the authors have answered makes if difficult to focus, both on the letter, no different colour, as in the text (light blue…)

It must also be defined on abstract…

“Although “EU” is s known abbreviation, consider defining it

Thank you. We have defined it now at the beginning of the introduction.

EKLIPSE is now defined in the introduction, as well as ALTER-Net. If the editor prefers that we also define them in the abstract we are ready to do so as well.

Some journals do not accept it

“Do not use “We”

We considered this suggestion but as the MICESE process was a very dynamic and active process, we felt this was better reflected with the use of “we”.

Content in “” IS a repetition of the text so that the authors are aware to what am I reefing to…

It was related to the “We” use…

“Consider entirely revising language according to suggested structure…

“ We provide insights and analyses of the 33 new method through which these key messages were generated and processed. We finally reflect 34 on how to improve the future involvement of scientists in science-policy interfaces.”

Unfortunately we are unsure what the reviewer is suggesting as the above sentence seems to be a repetition of the two last sentences of our abstract.

Although common practice in some journals before, it is not the practice everywhere now. No added knowledge

Consider rewriting:

These two sentences are indeed in the abstract and in the text itself. According to our experience, it is common practice in research that some sentences or parts of sentences are repeated in the abstract and the main text.

I do not agree with the authors. The language of the text could be improved. It is a report, rather than a scientific text.

If we were to replace “we”, we would be less assertive (and readable). We have therefore kept this sentence.

Conte was a typo and means “content”

I believe authors are not open to change:

“At this stage, only general purpose should be addressed, consider moving other content to abstract, discussion, conclusions, implications or limitations…

We do not know which part of the manuscript this refers to so we are afraid we cannot take it into account. We also do not know what “conte” refers to.

I am asking you to do that…

The guidelines for authors of Sustainability provide no guidance on footnotes but if the Editor requests there removal we will provide the information contained in them in a different way.

As I try to be open, the authors should too. This remark refers to previous comments, considering the style of the text. We all learn everyday…

“Understanding the intention, I have difficulty in seeing what is really relevant here…

We are sorry but we do not understand what this remark refers to in our paper.

It may not be in guidelines; it is a suggestion from someone also with experience

We have no experience of such a structure for the conclusion section and this is not referred to in the Sustainability guidelines. We have therefore kept the structure of the conclusion as it is.

This means that all comments expressed here must be considered and somewhat applied when analysing supplementary material.

I do not feel the need to state again that I know EKLIPSE well, and that the main objection has to do with the style and structure of the manuscript.

I personally feel that the “we” use “penalizes” the text, opposite to what was expressed by the authors.

Some additional comments.

Abbreviations should not be in captions…

“Figure 1. Schematic timeline representation of the MICESE process. MICESE stands for 183 Multiphased, Iterative and Consultative Elicitation of Scientific Expertise.”

Check “because” twice

“because it referred to geodiversity 190 which we perceived had no link through public policies and societal changes with biodiversity - 191 and five were merged with other KMs because of high overlap between them,”

Why “,” and not “-“ in values in Table 2?

Consider revising “Firstly,” the “listing” style.

Again, conclusions should start with a brief contextualization to justify the relevance of this publication.  I maintain previous comments

Some proofreading is necessary “We expect our work on the identification of Key Messages will”

Comments are intended to improve the text, nothing more

Author Response

We thank the reviewer for the effort to review a second version of our manuscript.

Comments and Suggestions for Authors

Highlight changes in yellow in a next revision, please. No track changes.

Thank you for the specification. We have followed this advice.

Consider comments in the entire text.

The way the authors have answered makes if difficult to focus, both on the letter, no different colour, as in the text (light blue…)

We are sorry for this: we asked the office of the journal which version we should post and they answered us the version with modifications, which is why we addressed the review in this way.

It must also be defined on abstract…

“Although “EU” is s known abbreviation, consider defining it

Thank you very much for the suggestion. We have defined it now at the beginning of the introduction.

EKLIPSE is now defined in the introduction, as well as ALTER-Net. If the editor prefers that we also define them in the abstract we are ready to do so as well.

            We agree. We have specified what EU meant in the abstract and define  ALTER-Net and EKLIPSE in the introduction in order to keep the abstract brief and snappy.

Some journals do not accept it

“Do not use “We”

We considered this suggestion but as the MICESE process was a very dynamic and active process, we felt this was better reflected with the use of “we”.

We acknowledge that the reviewer maintains his/her point of view and assume that this is linked to the use of the passive voice in certain scientific papers. Our experience, however, is that the use of the active voice is very much a question of preference, and increasingly, journals are encouraging the use of the active voice. In this paper, we feel that the active voice makes the paper much more dynamic and easier to read generally. We would therefore prefer to keep the use of “we”, especially as the MICESE approach was a very active process. As such, the voice used in the paper reflects the active process of the MICESE approach. We made an exception on this choice: the Material & Methods section was turned to the passive voice - at least avoiding any “we” - as it requires a more neutral tone than the other sections.

Content in “” IS a repetition of the text so that the authors are aware to what am I reefing to…

It was related to the “We” use…

“Consider entirely revising language according to suggested structure…

“ We provide insights and analyses of the 33 new method through which these key messages were generated and processed. We finally reflect 34 on how to improve the future involvement of scientists in science-policy interfaces.”

Unfortunately we are unsure what the reviewer is suggesting as the above sentence seems to be a repetition of the two last sentences of our abstract.

            In our experience, it is common to repeat elements in the abstract, as the abstract is a summary of the overall paper. In terms of the use of “we”, see our above justification.  .

Although common practice in some journals before, it is not the practice everywhere now. No added knowledge

Consider rewriting:

These two sentences are indeed in the abstract and in the text itself. According to our experience, it is common practice in research that some sentences or parts of sentences are repeated in the abstract and the main text.

We were somewhat surprised by this comment. Indeed, there  is no added knowledge in an abstract as the latter  is a synthesis of the knowledge in the paper.

I do not agree with the authors. The language of the text could be improved. It is a report, rather than a scientific text.

If we were to replace “we”, we would be less assertive (and readable). We have therefore kept this sentence.

The two other reviewers in round 1 seemed to interpret the paper as such, and not as a scientific report. Indeed there is no scientific report on which this paper is based. Perhaps it is the scientific messages that come across as a scientific report, but they are the compiled and jointly developed key messages from the scientific community. The paper has been proofread by two native British speakers. .

Conte was a typo and means “content”

I believe authors are not open to change:

“At this stage, only general purpose should be addressed, consider moving other content to abstract, discussion, conclusions, implications or limitations…

We do not know which part of the manuscript this refers to so we are afraid we cannot take it into account. We also do not know what “conte” refers to.

We have made every effort to take into account the constructive suggestions of the reviewers. Indeed, from a quick analysis of the response to the two other reviewers in stage 1:

  • Reviewer 1, pdf file: all comments were taken into account except 2 (with explanations why)
  • Reviewer 1, explicit list of comments: we modified the text along the spirit of the 4 comments
  • Reviewer 3: we have taken into account 30 of the 32 comments received.

These are good indicators that we are not as narrow minded as the reviewer suggests we are. On the contrary, we are extremely grateful for the suggestions made by the reviewers and believe these have greatly improved the manuscript. As with all peer reviewing processes, it is an iterative process that aims to improve the quality of the papers published. We therefore drew on these reviews in this spirit of improvement.

I am asking you to do that…

The guidelines for authors of Sustainability provide no guidance on footnotes but if the Editor requests there removal we will provide the information contained in them in a different way.

It is unclear if the reviewer is an Editor. We wait for a clear message from an Editor of Sustainability on whether to remove these footnotes. In our view, having the footnotes integrated in the main text will be confusing for the reader.

As I try to be open, the authors should too. This remark refers to previous comments, considering the style of the text. We all learn everyday…

“Understanding the intention, I have difficulty in seeing what is really relevant here…

We are sorry but we do not understand what this remark refers to in our paper.

We have endeavoured to highlight the relevance of the paper throughout the text. We aim to be as open to reviewers as we can - hence our inclusion of the reviewer feedback in this new version of the manuscript (also see above for a clear run-through of all the comments integrated).

It may not be in guidelines; it is a suggestion from someone also with experience

We have no experience of such a structure for the conclusion section and this is not referred to in the Sustainability guidelines. We have therefore kept the structure of the conclusion as it is.

We considered the structure of the conclusion but realised that there was then a risk of repetition hence the decision, based on the feedback from the other two reviewers, to keep the same structure, but adding some extra elements to bring out the relevance of the paper, as suggested by this reviewer.

This means that all comments expressed here must be considered and somewhat applied when analysing supplementary material.

 I do not feel the need to state again that I know EKLIPSE well, and that the main objection has to do with the style and structure of the manuscript.

 I personally feel that the “we” use “penalizes” the text, opposite to what was expressed by the authors.

We feel that the use of the active voice helps the reader, but also helps reflects the active MICESE process. We are not sure how the use of “we” can penalise the text (except in the M&M section where we judged is was not improving the lanuscript). Perhaps this is the reference to the text sounding like a scientific report? Although a number of scientific reports also use a passive voice. Again, perhaps this is a matter of preference, but in this case the text seemed much heavier and less accessible with the passive voice rather than the use of “we”.

Some additional comments.

Abbreviations should not be in captions…

“Figure 1. Schematic timeline representation of the MICESE process. MICESE stands for 183 Multiphased, Iterative and Consultative Elicitation of Scientific Expertise.”

We have taken this into account and added the explanation of the abbreviation  immediately after its use.  

Check “because” twice

“because it referred to geodiversity 190 which we perceived had no link through public policies and societal changes with biodiversity - 191 and five were merged with other KMs because of high overlap between them,”

            Thank you for this clear comment. We have replaced the second “because”.

Why “,” and not “-“ in values in Table 2?

We guess the reviewer refers to the decimal point (“.”)  instead of “-”. We have done this replacement.

Consider revising “Firstly,” the “listing” style.

            We agree with this excellent suggestion and have made the change.

Again, conclusions should start with a brief contextualization to justify the relevance of this publication.  I maintain previous comments

Many thanks for this suggestion - we have now incorporated a first sentence in the conclusion as contextualisation.

Some proofreading is necessary “We expect our work on the identification of Key Messages will”

            The manuscript has been read by two native British speakers.

Comments are intended to improve the text, nothing more

We greatly appreciate your intent and indeed the peer reviewing process carried out as part of this submission process. It seems that we have accepted most of your suggestions, but that the main issue now remains in terms of the use of “we”. We believe that this approach is reflected in many journals, which increasingly call for the active voice in the text, and perhaps more so in this paper, where we describe an active process.